# Pharmacokinetic and Pharmacogenetic Predictors of Major Bleeding Events in Patients with an Acute Coronary Syndrome and Atrial Fibrillation Receiving Combined Antithrombotic Therapy

**DOI:** 10.3390/jpm13091371

**Published:** 2023-09-12

**Authors:** Olga Baturina, Maria Chashkina, Denis Andreev, Karin Mirzaev, Alexandra Bykova, Alexandr Suvorov, Daria Yeryshova, Svetlana Suchkova, Dmitry Sychev, Abram Syrkin

**Affiliations:** 1Cardiology, Functional and Ultrasound Diagnostics Department, N.V. Sklifosovskiy Institute of Clinical Medicine, Sechenov First Moscow State Medical University, Ministry of Health of the Russian Federation, Moscow 119048, Russia; o.a.baturina@ya.ru (O.B.); dennan@mail.ru (D.A.); bykova_a_a@staff.sechenov.ru (A.B.); darya.eryshova@yandex.ru (D.Y.); lanasuchkova@mail.ru (S.S.); previntenscardiology@yandex.ru (A.S.); 2Clinical Pharmacology and Therapy Department B.E. Votchal, Russian Medical Academy of Continuous Professional Education, Ministry of Health of the Russian Federation, Moscow 125993, Russia; karin05doc@yandex.ru (K.M.); dimasychev@mail.ru (D.S.); 3World-Class Research Center “Digital Biodesign and Personalized Healthcare”, Sechenov First Moscow State Medical University, Ministry of Health of the Russian Federation, Moscow 119048, Russia; suvorov_a_yu_1@staff.sechenov.ru

**Keywords:** combined antithrombotic therapy, rivaroxaban, clopidogrel, dual antiplatelet therapy, triple antiplatelet therapy, atrial fibrillation, acute coronary syndrome, major bleeding

## Abstract

**Objective:** This study’s objective was to evaluate the effects of pharmacokinetic and pharmacogenetic factors on major bleeding in patients with ACS and non-valvular AF receiving combined antithrombotic therapy consisting of rivaroxaban, clopidogrel, and aspirin as part of dual or triple therapy. **Methods:** A prospective observational study was conducted in two PCI centers in Moscow, the Russian Federation, from 2017 to 2018. One hundred patients with ACS and AF were enrolled. Prospective follow-ups continued for 12 months. **Results:** A total of 36 patients experienced bleeding events, with 10 experiencing major bleeding based on the BARC scale and 17 experiencing major bleeding based on the ISTH scale. The following predictors associated with an increased number of major bleeding events were identified: for the ISTH scale, a Css min. of rivaroxaban of >137 pg/mL (5.94 OR, (95% CI, 3.13–12.99; *p* < 0.004)) and carriage of the T allelic variant polymorphism ABCB1 rs4148738 (8.97 OR (95% CI, 1.48–14.49; *p* < 0.017)), as well as for the BARC scale (5.76 OR (95% CI, 2.36–9.87; *p* < 0.018)). **Conclusions:** Measuring residual steady-state rivaroxaban concentrations and determining the carriage of the T allelic variant polymorphism ABCB1 rs4148738 may be applicable to high-risk patients for subsequent antithrombotic therapy modification.

## 1. Introduction

Selecting the most suitable antithrombotic regimen can be a complex task, given that approximately 7–15% of patients who have acute coronary syndrome (ACS) also have concomitant atrial fibrillation (AF) [1,2].

Treatment optimization for patients with AF, ACS, and/or percutaneous coronary intervention (PCI) is being advanced further with the use of randomized clinical trials (RCTs) involving direct oral anticoagulants (DOACs) such as PIO-NEER AF PCI (rivaroxaban) [3], RE-DUAL PCI (dabigatran) [4], and AUGUSTUS (apixaban) [5], studies on which have been performed in large groups of patients. The results of these RCTs show that using DOACs in patients with AF, ACS, and/or PCI significantly reduces the incidence of bleeding events compared with the anticoagulant warfarin. This confirms the WOEST study’s [6] findings on the safe and effective use of combined antithrombotic therapy (anticoagulant and antiplatelet agent) compared with prolonged triple antithrombotic therapy (anticoagulant and dual antithrombotic therapy (DAT)).

In line with current guidelines, patients with ACS and AF undergoing PCI are advised to receive a combination of DOAC and DAT for 1 week. In case of a high risk of recurrent ischemic events, triple therapy may be continued for up to 1 month [7,8,9,10]. For patients on medical treatment, dual therapy is recommended for a duration of up to 12 months. Among thienopyridines, clopidogrel is recommended as a cotreatment due to the limited number of patients in RCTs who receive ticagrelol or prasugrel as part of triple (TAT) and dual therapies [3].

The biotransformation of clopidogrel can be influenced by genetic factors such as the carriage of alleles with a reduced functional activity of CYP2C19 (CYP2C19*2 and *3) and ABCB1 P-glycoprotein polymorphisms [11,12,13,14,15,16,17,18]. Trials have observed variability in responses to clopidogrel, which is attributed to various factors, including the CYP2C19-dependent production of active metabolites [18,19]. Data show that individuals who possess the CYP2C19*17 gene variant are more susceptible to experiencing bleeding complications while undergoing treatment with clopidogrel [19]. Furthermore, published information suggests that individuals with the ABCB1 CT and TT genotypes are significantly associated with a higher risk of bleeding [20,21].

To reduce the risk of bleeding, in addition to influencing modifiable risk factors for hemorrhagic complications, an assessment of the pharmacokinetic parameters of DOACs has been proposed, including polymorphisms in certain genes involved in the absorption, biotransformation, and elimination of DOACs, which can potentially influence the pharmacokinetics of these medications. It has been proven that the metabolism of dabigatran involves the CES1, CES2, and ABCB1 genes [22]. When assessing the influence of the ABCB1 gene’s rs4148738 variant, it was found that carriers of the AA genotype had significantly higher maximum steady-state concentration of apixaban values compared with carriers of the G allele [23]. It has been established that the ABCB1, ABCG2, CYP3A4, CYP3A5, and CYP2J2 genes are involved in the metabolism of rivaroxaban [24,25,26].

A widely utilized tool for evaluating the bleeding risk in patients with AF is the HAS-BLED scale, which aims to quantitatively assess the bleeding risk associated with anticoagulant use. However, despite its extensive use, the applicability of this scoring system in making optimal decisions about antithrombotic therapy for AF patients with indications for combined antithrombotic therapy (ATT) remains uncertain. It is crucial to evaluate which risk prediction tools can consider specific patient characteristics and aid healthcare providers in selecting the most suitable antithrombotic regimen for each case. Therefore, additional factors that could potentially impact bleeding events in this specific patient population must be explored.

**Study objective**: This study’s objective was to evaluate the effect of pharmacokinetic and pharmacogenetic factors on major bleeding according to the BARC and ISTH scales in patients diagnosed with ACS and non-valvular AF receiving combined antithrombotic therapy consisting of rivaroxaban, clopidogrel, and aspirin as part of dual or triple antithrombotic therapy.

## 2. Materials and Methods

**Study setting and duration.** This prospective observational study was conducted at the N.I. Pirogov City Clinical Hospital No. 1 and the S.S. Yudin City Clinical Hospital in Moscow, the Russian Federation, from 1 October 2017 to 30 November 2018.

**Inclusion and exclusion criteria.** The inclusion criteria were as follows: over 18 years of age; presenting with myocardial infarction (MI)/unstable angina (UA) upon current admission; presenting with non-valvular AF confirmed with an ECG; and receiving combined ATT with rivaroxaban as an anticoagulant and clopidogrel only or clopidogrel in combination with aspirin as an antiplatelet agent.

The exclusion criteria were as follows: pregnancy and lactation; active internal bleeding; liver cirrhosis with hepatic insufficiency class C according to the Child–Pugh classification; stage 5 chronic kidney disease; HIV infection; alcohol/drug addiction; moderate or severe mitral valve stenosis; mechanical heart valves; severe mental disorders; allergic reactions/drug intolerances; a patient’s unwillingness to participate in this study.

This study enrolled patients with MI based on the fourth universal definition of MI. Unstable angina was diagnosed when criteria for myocardial ischemia at rest or upon minimal exertion were present without evidence of myocardial necrosis markers. Initial diagnoses were carried out by the visiting physicians at the hospitals, and final diagnoses were documented in the discharge summary.

**Follow-up procedures.** Following discharge from the hospitals, prospective follow-up continued for 12 months, which included patients with AF and ACS who did not meet the exclusion criteria. The follow-up involved telephone or in-person contact with the patients at the following intervals: one month after inclusion and then at three, six, nine, and twelve months. During the visits, adherence to antiplatelet therapy was assessed using the Morisky–Green questionnaire, and standard laboratory tests were conducted 6 months after discharge.

**Pharmacological testing and platelet reactivity.** Pharmacological testing was also performed. Platelet reactivity was assessed using a portable VerifyNow test system (Accumetrics, USA) within 2 h after whole venous blood samples were drawn into 2 mL vacuum tubes containing 3.2% sodium citrate and kept at a temperature range of 2−25 °C. Blood sampling was performed 3–7 days after a patient developed acute coronary syndrome (ACS). The obtained residual platelet reactivity (RPR) is expressed in arbitrary units, PRU (P2Y12 reaction units). The manufacturers recommended using threshold values to interpret the results, with the maximum being PRU = 208 and the minimum being PRU = 95.

The quantitative determination of rivaroxaban in blood plasma samples was carried out using the high-performance liquid chromatography method coupled with mass spectrometry according to the established procedure. Blood was collected into vacuum tubes containing 6 mL of lithium heparin 4–5 days after a complete blood count was prescribed before the patient took their next dose of rivaroxaban. The tubes were centrifuged at 15 thousand rpm for 15 min to obtain the plasma. The obtained plasma was aliquoted and stored in Eppendorf tubes at −60 °C until further analysis. This study utilized the following equipment: an Agilent 1200 HPLC System, an Agilent Extend-C18 column, and an Agilent Triple Quad LC/MS 6410 mass spectrometer. The mass spectrum of rivaroxaban was registered using the multiple reaction monitoring (MRM) mode. Threshold values for the minimum steady-state rivaroxaban concentration (Css min.) were defined based on previous research results and EHRA recommendations.

**Genotyping procedure.** For genotyping, venous blood was collected into vacuum tubes containing ethylenediaminetetraacetic acid (EDTA) 3–7 days after acute coronary syndrome (ACS) occurred. The venous blood tubes were frozen at −60 °C until the genotyping procedure was conducted. The identification of polymorphic gene markers’ carriage of ABCB1, CYP3A53, CYP3A422, CYP2C192, CYP2C193, and CYP2C19*17 was carried out using the real-time polymerase chain reaction (real-time PCR) method.

**Statistical processing and analysis.** Statistical processing was performed using the Python v3.8 programming language. For quantitative indicators, the nature of the distribution was determined using the Shapiro–Wilk test, including the mean value, standard deviation, median, interquartile range, 95% confidence interval, and minimum and maximum values. The proportional and absolute value counts were determined for categorical and qualitative features. A comparative analysis of normally distributed quantitative features was conducted based on Welch’s *t*-test (for two groups) or ANOVA (for more than two groups), followed by pairwise group comparisons. When quantitative features were not normally distributed, the analysis was performed using the Mann–Whitney U test (for two groups) or the Kruskal–Wallis test (for more than two groups). A comparative analysis of categorical and qualitative features was carried out using Fisher’s exact test. Regression analysis was used by constructing equations involving one variable to estimate the effects of factors on outcomes. Significant variables were included in a single multiple regression equation to find independent endpoint predictors. Several real significant variables were converted to binary ones, identifying the optimal threshold from the Youden index during the ROC analysis. The significance level for the comparative and regression analyses was 0.05.

## 3. Results

The study involved 100 patients. The group’s median age was 74 years [64;81]. Among the participants, 26% experienced MI with ST elevation, 51% had MI without ST elevation, and 23% presented with UA. The prevalent comorbidities in this cohort were hypertension and obesity, as shown in Table 1.

All patients received ATT, including rivaroxaban. Among them, 27 patients (27%) were prescribed DAT consisting of rivaroxaban and clopidogrel, while 73 patients (73%) were prescribed TAT comprising rivaroxaban, clopidogrel, and aspirin.

Regarding the specific dosage of rivaroxaban, 20 mg was prescribed to four patients on TAT and three patients on DAT. A dose of 15 mg was administered to 66 patients on TAT and 24 patients on DAT. A dose of 10 mg was recommended solely for patients on TAT and given to three patients. At discharge, patients receiving DAT continued the therapy for 12 months.

The duration of TAT, ranging from 1 to 12 months, varied among the patients, as shown by the following:One month—n = 48;Three months—n = 9;Six months—n = 14;Twelve months—n = 2.

Figure 1 presents the distribution of the duration of TAT depending on the CHA2DS2-VASc score.

### 3.1. TAT—Triple Antithrombotic Therapy

Thirty-eight patients experienced bleeding during the 12 months of observation. Table 2 and Figure 2 depict the distribution of bleeding based on the source of bleeding.

Figure 3 illustrates the distribution of bleeding occurrences in patients within 12 months, categorized by severity and based on the chosen scale. As depicted in Figure 3, the severity of bleeding measured with different scales (BARC and ISTH) significantly differed (*p* < 0.0001). By the end of the observation period, 10 patients had major bleeding according to the BARC scale, and 17 patients had major bleeding according to the ISTH scale (Figure 4). No fatal bleeding events were observed.

Figure 5 shows the Kaplan–Meier curve depicting the proportion of patients who had bleeding events during the entire follow-up period (12 months). Half of all hemorrhagic events occurred in the first two months after the start of combined ATT.

When considering major bleeding events based on the composition of the combined ATT, our analysis reveals that, according to the BARC scale, there were four major bleeding occurrences in the DAT group and six in the TAT group. Similarly, following the ISTH scale, there were 6 major bleeding events in the DAT group and 11 in the TAT group. Figure 6 provides a visual representation of the distribution of major bleedings for both the TAT and DAT groups.

### 3.2. DAT—Dual Antithrombotic Therapy, TAT—Triple Antithrombotic Therapy

Additionally, the patients included in the analysis were investigated for gene polymorphisms encoding proteins involved in the metabolism of antithrombotic medications, the minimum steady-state rivaroxaban concentration in plasma (Css min), RPR (PRU), and the % of platelet inhibition.

The distributions of the ABCB1, CYP2C19*2, CYP2C19*17, and CYP3A5*3 genotypes followed the Heidi–Weinberg principle (Table 3). The CYP2C19*3 and CYP3A4*22 C>T rs35599367 polymorphisms were not included in the subsequent analysis.

A comparative analysis of patients with and without bleeding is presented in Table 4. As evident in Table 4, patients with bleeding had more severe coronary artery disease according to the SYNTAX scores (*p* = 0.005), and a higher incidence of stage 3a CKD was observed in this group (*p* = 0.35). Paradoxically, the proportion of patients with diabetes was higher in the group without bleeding, which was likely due to a statistical error. Apart from these characteristics, both groups were comparable. It is worth noting that there were no differences between the groups concerning HASBLED scores.

Considering the indirect impact of rivaroxaban on platelet activity in addition to using the standard PRU 95–208 values for clopidogrel therapy, we explored a more pertinent cut-off point for RPR that could apply to the specific patient population under investigation. To perform logistic regression using ROC analysis, we determined the threshold value of 159 for RPR, with an AUC of 76%, sensitivity of 70.6%, and specificity of 69.9.

Subsequently, both univariate and multivariate analyses were conducted on the bleeding scales—BARC and ISTH—with adjustments for gender and age.

During the univariate analysis, it was observed that the probability of hemorrhagic complications (assessed using the BARC scale) increased in the presence of the following factors (Table 5): carriage of the CT and TT genotypes of ABCB1 rs4148738 as well as the CC genotype of ABCB1 3435 and RPR (with a PRU of less than 159 pg/mL).

According to the ISTH scale, the following factors associated with a higher incidence of bleeding events were identified (Table 6): a Css min. of rivaroxaban of more than 137 ng/mL, the TT genotype of ABCB1 rs4148738, the CC genotype of ABCB1 3435, and residual platelet reactivity (with a PRU of less than 159 pg/mL).

The multivariate analysis identified the following factors associated with an increased number of major bleeding events:For the ISTH scale:-A higher rivaroxaban concentration and Css min. greater than 137 pg/mL (5.94 OR (95% CI, 3.13–12.99; *p* < 0.004);-Carriage of the T allelic variant polymorphism ABCB1 rs4148738 (8.97 OR (95% CI, 1.48–14.49; *p* < 0.017).For the BARC scale:
-The influence of the T allelic variant polymorphism ABCB1 rs4148738 (5.76 OR (95% CI, 2.36–9.87; *p* < 0.018).

Other factors, such as RPR (PRU < 159) and other genetic polymorphisms, did not demonstrate a significant impact on the occurrence rate of major bleeding events.

## 4. Discussion

Currently, clinical practice adopts a “bleeding avoidance” strategy to prevent hemorrhagic events. A crucial aspect of this approach involves the development of individualized antiplatelet therapy tactics based on bleeding risk assessment, for which numerous scoring systems have been devised [7]. Nevertheless, their diagnostic value may substantially vary across different patient cohorts, as prognostic scales do not consider the use of combined antithrombotic therapy, including anticoagulants. The findings of our study demonstrate that the HAS-BLED score exhibits limited prognostic value when applied to a population of patients receiving combined antithrombotic therapy. Thus, searching for bleeding predictors in this specific patient population remains pertinent.

The measurement of DOAC concentrations is becoming an integral part of routine clinical practice, especially for patients undergoing combined ATT. This advancement allows for personalized treatment approaches and the consideration of individual pharmacokinetic profiles. This strategy not only minimizes adverse events but also optimizes therapeutic outcomes, effectively managing the balance between bleeding risks and the prevention of thromboembolic events, a particularly important consideration for patients with AF.

While no large-scale study of rivaroxaban plasma concentrations in patients with atrial fibrillation was conducted, we determined the maximum threshold value of Css min. to be 137 ng/mL and the minimum value to be 12 ng/mL. None of the patients in our study exhibited rivaroxaban Css levels below the minimum threshold (<12 ng/mL). Approximately 13% of the patients exceeded the concentration threshold (>137 ng/mL). The median Css min. in our study was 58 [39;99], with the highest measured plasma Css min. being 282 ng/mL. These findings are marginally higher than the median rivaroxaban concentrations reported by Al-Aieshy et al., where the minimum concentration was 34 ng/mL [27]. In a study conducted by Sennesael et al., the Css min. in 10 patients receiving rivaroxaban at trough were between 12 and 251 ng/mL (with a median value of 94 ng/mL) [28]. Previous studies have demonstrated substantial variability in rivaroxaban concentration levels, ranging from 8 to 660 ng/mL.

A higher concentration of the medication was associated with an increased frequency of major bleeding events, as measured using the ISTH scale (5.94 OR; 95% CI, 3.13–12.99; *p* < 0.004). These findings align with the results of a study conducted on 60 patients receiving rivaroxaban at doses of 20 mg and 15 mg daily, which demonstrated a significant correlation between steady-state rivaroxaban concentrations and bleeding frequency in patients with atrial fibrillation (48 ± 30 vs. 34 ± 26; *p* = 0.02) [29]. Moreover, as per two cohort studies, patients who experienced severe bleeding events had a median rivaroxaban level of 124 ng/mL, while those admitted for intracranial hemorrhage had a median rivaroxaban level of 102 ng/mL [30,31].

The absorption and elimination of rivaroxaban are primarily influenced by the P-glycoprotein (P-gp) encoded by ABCB1 and BCRP (encoded by the ABCG2 gene) [24]. According to our study’s results, carriage of the TT allelic variant polymorphism ABCB1 rs4148738 is associated with an increased frequency of major bleeding events, as evidenced by both the BARC scales (5.76 OR; 95% CI, 2.36–9.87; *p* < 0.018) and the ISTH scales (8.97 OR; 95% CI, 1.48–14.49; *p* < 0.017). In several investigations of patients receiving rivaroxaban therapy, ABCB1 gene polymorphisms influenced the plasma concentration of rivaroxaban, but they were not linked to an elevated frequency of major bleeding events [32,33]. Nevertheless, other published studies have reported results consistent with our findings. Sennesael et al. reported that three out of four patients with major bleeding and rivaroxaban concentrations above the expected level (greater than 136 ng/mL) were heterozygous for 1236C>T, 2677G>T, 3435C>T, and rs4148738 ABCB1 [28]. Subsequent studies also indicated that carriage of 2677G>T and 3435C>T ABCB1 was associated with higher plasma concentrations of rivaroxaban and hemorrhagic complications [21].

Upon completion of the observation period, major bleeding events were observed in 10 patients as per the BARC scale and 17 patients according to the ISTH scale. Such discrepancies in the number of hemorrhagic events can be attributed to the variations in bleeding categorization between the two scoring systems. Similar differences in the incidence of bleeding events, depending on the utilized scoring system, have been described in a sub-analysis of the ENGAGE AF-TIMI 48 trial [34], which compared the efficacy and safety of edoxaban and warfarin in patients with atrial fibrillation. Various bleeding scales, including ISTH, TIMI, GUSTO, and BARC, were compared in this analysis. The results demonstrated that the more graded distribution provided by the ISTH scale could help differentiate the entire spectrum of bleeding, while the BARC scale could aid in identifying more severe cases of bleeding.

In the context of patients with AF undergoing PCI, the significance of TAT becomes notably pronounced. As highlighted by Golwala et al. [35], the debate surrounding DAT and TAT effectiveness continues, with DAT being proposed as a safer alternative due to its potential reduction in bleeding risk. However, we concur with the perspective that the choice between DAT and TAT requires careful consideration, and the conclusive notion that DAT is universally superior might not be applicable to all patients with AF post-PCI.

The optimal duration and composition of TAT remain subjects of active investigation, as demonstrated by the variability in therapy duration across studies [3,4,5]. The ongoing MASTER DAPT study [36], evaluating the safety and efficacy of different TAT durations while allowing for the use of DOACs, aims to provide further insights into refining antithrombotic strategies for this patient population. In summary, the choice of antithrombotic therapy for patients with AF undergoing PCI is a multifaceted one, with triple antithrombotic therapy remaining an essential consideration. While DAT may offer potential advantages in terms of bleeding risk reduction, the broader clinical context, patient characteristics, and ongoing research efforts collectively contribute to shaping the optimal therapeutic approach.

A subset of patients in our study received TAT for 3 months (n = 14) and 6 months (n = 2). These decisions were based on the clinical judgment of hospitals and outpatient healthcare providers, considering their own experiences and individual patient characteristics. Prolonged TAT use did not show a significant association with the occurrence of major bleeding events in the multivariate analysis.


**Study limitations.**


This study was conducted exclusively at two medical centers in Russia, which may restrict the generalizability of the findings to a broader population. Additionally, the exclusive use of rivaroxaban as an anticoagulant limits the applicability of the results to other DOACs. Moreover, the small sample size of patients should also be recognized as a limitation of this study. To extrapolate the findings to the patient population with ACS and AF, larger-scale studies involving more substantial patient cohorts are required.

## 5. Conclusions

A higher concentration of rivaroxaban (Css min. > 137 ng/mL) and carriage of the T allelic variant polymorphism ABCB1 rs4148738 are associated with an increased incidence of major bleeding events in patients with ACS and AF who are receiving combined antithrombotic therapy comprising rivaroxaban and clopidogrel. This study’s findings may be applicable to high-risk patients for subsequent therapy modification.

## Figures and Tables

**Figure 1 jpm-13-01371-f001:**
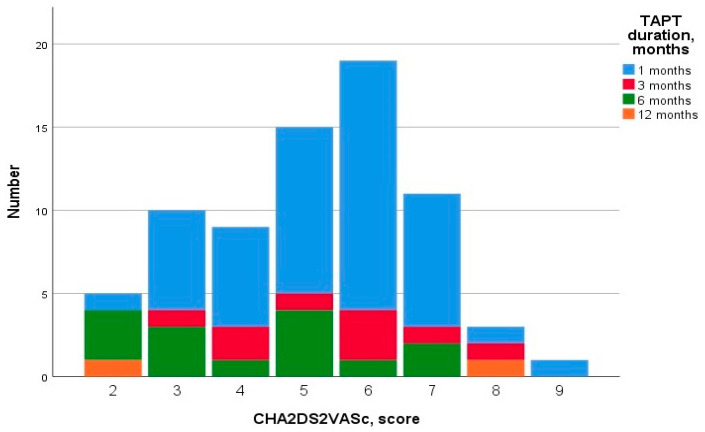
Duration of TAT depending on the CHA2DS2-VASc score.

**Figure 2 jpm-13-01371-f002:**
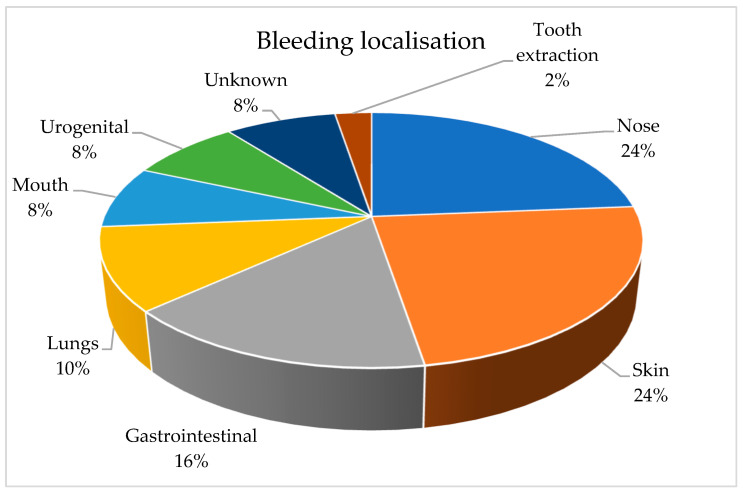
Distribution of bleeding depending on bleeding localization.

**Figure 3 jpm-13-01371-f003:**
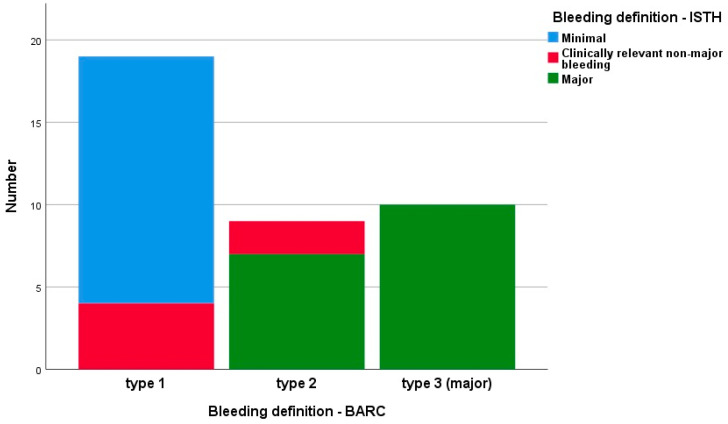
Distribution of bleeding occurrences categorized by severity based on the BARC and ISTH scales (n = 38).

**Figure 4 jpm-13-01371-f004:**
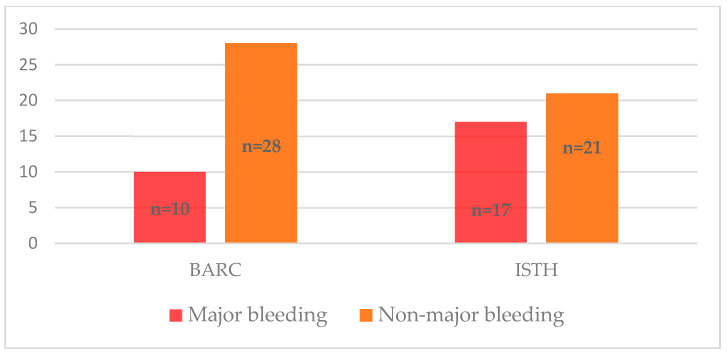
Major and non-major bleedings according to the BARC and ISTH scales.

**Figure 5 jpm-13-01371-f005:**
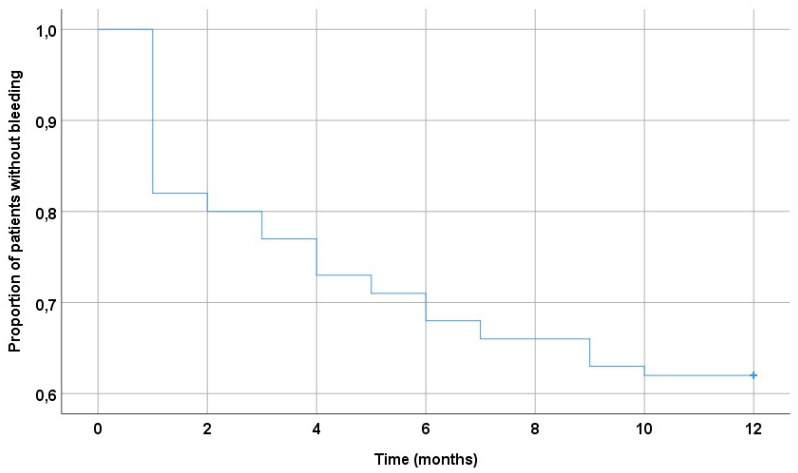
Kaplan–Meier curve for bleeding in patients during the entire follow-up period.

**Figure 6 jpm-13-01371-f006:**
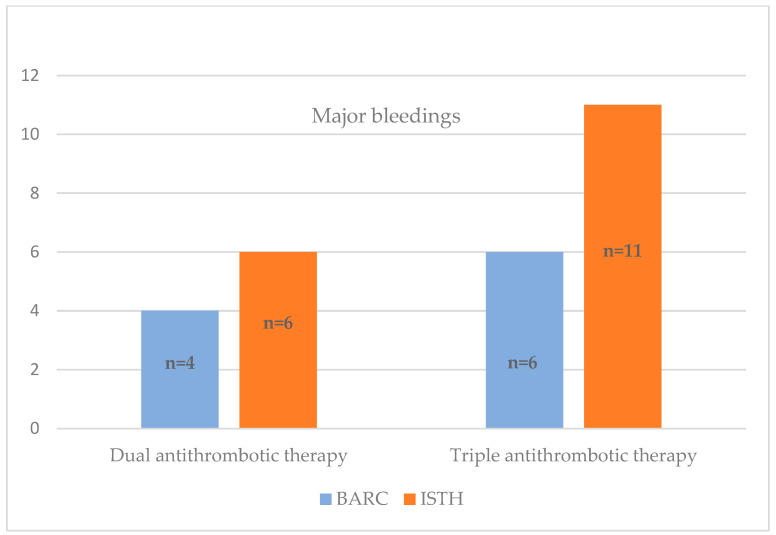
Distribution of major bleeding occurrences for DAT and TAT according to the BARC and ISTH scales.

**Table 1 jpm-13-01371-t001:** Patients’ demographics and clinical characteristics.

Characteristics	Patients with MI/UA and AF, n = 100
Age, years, Me [25;75]	74 [64;81]
Men/women, n (%)	56/44; (56/44)
Active smoking, n (%)	12 (12)
ST-segment elevation myocardial infarction/non-ST-elevation myocardial infarction Unstable angina	27/51 (27/51) 22 (22)
Percutaneous coronary intervention at current admission, n (%)	73 (73)
CRUSADE score, M ± SD GRACE score, M ± SD Acute heart failure, KILLIP score: Class I–II, n (%) Class III–IV, n (%)	43 ± 13 173 ± 34 91/(91) 9/(9)
Concomitant and previous diseases: Hypertension, n (%) Obesity, n (%) Diabetes, n (%) Coronary revascularization, n (%) Chronic heart failure, n (%) Acute cerebrovascular event, n (%) Peripheral arterial disease, n (%) Chronic obstructive lung disease, n (%) Active cancer, n (%) Erosive and ulcerative diseases of the gastrointestinal tract, n (%) History of major bleedings, n (%) Anemia at discharge, n (%): ● Mild (129/119–110 g/dL), n (%) ● Moderate (109–80 g/dL), n (%) ● Severe (<80 g/dL), n (%) Chronic kidney disease: ● Stage 3Aa, n (%) ● Stages 3B–4, n (%) ● Stage 5, n (%)	99 (99) 53 (53) 31 (31) 36 (36) 34 (34) 17 (17) 26 (26) 22 (22) 2 (2) 11 (11) 11 (11) 32 (32) 24 (24) 6 (6) 2 (2) 37 (37) 23 (23) 5 (5)
CHA2DS2-VASc, Me [25;75]	5 [4;6]
HAS-BLED, Me [25;75]	2.5 [2;3]

**Table 2 jpm-13-01371-t002:** Distribution of bleeding depending on localization.

Bleeding Point	N	% (n = 38)
Nose	9	23.7
Skin	9	23.7
Gastrointestinal tract	6	15.8
Lungs	4	10.5
Bleeding in the mouth	3	7.9
Urogenital	3	7.9
Source unknown	3	7.8
At tooth extraction	1	2.6

**Table 3 jpm-13-01371-t003:** Hardy–Weinberg equilibrium test.

Genetic Polymorphism	Genotype	n	%	χ2
CYP3A4*22 C>T rs 35599367	CC CT	99 1	99.0 1.0	0.003
CYP3A5*3 rs776746	GG AG	82 18	82.0 18.0	0.978
ABCB1 rs418738	CC CT TT	18 45 37	18.0 45.0 37.0	0.532
ABCB1 rs1045642	CC CT TT	31 43 26	31.0 43.0 26.0	1.90
CYP2C19*2 rs4244285	AA GA GG	2 29 69	2.0 29.0 69.0	0.275
CYP2C19*17 rs12248560	CC CT TT	58 38 4	58.0 38.0 4.0	0.531
CYP2C19*3	GG	100	100.0	0

**Table 4 jpm-13-01371-t004:** Comparative characteristics of patients with and without bleeding events.

Characteristics	Without Bleeding (n = 62)	With Bleeding (n = 38)	*p*-Value
Age, years, Me [25;75]	70.5 [64;79.5]	77 [65;86]	0.057
Men/women, n (%)	35/27 (56.5/43.5)	21/17 (55.3/44.7)	NS
Active smoking, n (%)	8 (12.9)	4 (10.5)	NS
CHA2DS2-VASc score, Me [25;75]	5.0 [4;6]	5.5 [4;6]	NS
HAS-BLED score, Me [25;75]	2 [2;3]	3 [2;3]	NS
Acute coronary syndrome, n, %			
ST-segment elevation myocardial infarction	13 (21.0)	14 (36.8)	NS
Non-ST-elevation myocardial infarction	33 (53.2)	18 (47.4)	NS
Unstable angina	16 (25.8)	6 (15.8)	NS
CRUSADE score, M ± SD	34 ± 13.4	42.5 ± 12.2	NS
GRACE score, M ± SD	168 ± 34	180 ± 33	0.069
SYNTAX score, Me [25;75]	15 [9;24]	23.7 [14;36.4]	0.005 *
Acute heart failure, KILLIP score:			
Class I–II, n (%)	58 (93.5)	33 (86.8)	NS
Class III–IV, n (%)	4 (6.5)	5 (13.2)	NS
Concomitant and previous diseases			
Hypertension, n (%)	61 (98.4)	38 (100)	NS
Obesity, n (%)	36 (58.1)	17 (44.7)	NS
Body mass index, kg/cm^2^, Me [25;75]	30 ± 5.2	28.3 ± 4.2	0.087
Diabetes, n (%)	25 (40.3)	6 (15.8)	0.01 *
Coronary revascularization, n (%)	26 (42)	10 (26.4)	NS
Chronic heart failure, n (%)	20 (32.3)	14 (36.8)	NS
Acute cerebrovascular event, n (%)	9 (14.5)	8 (21.1)	NS
Peripheral arterial disease, n (%)	15 (24.2)	11 (28.9)	NS
Chronic obstructive lung disease, n (%)	14 (22.6)	8 (21.1)	NS
Active cancer, n (%)	0	2 (5.3)	0.068
Erosive and ulcerative diseases of the gastrointestinal tract, n (%)	6 (9.7)	5 (13.2)	NS
History of major bleedings, n (%)	5 (8.1)	6 (15.8)	NS
Anemia at discharge, n (%):	16 (25.8)	16 (42.1)	0.09
Mild (129/119–110 g/L), n (%)	11 (17.7)	13 (34.2)	0.061
Moderate (109–80 g/L), n (%)	4 (6.5)	2 (5.3)	NS
Severe (<80 g/L), n (%)	1 (1.6)	1 (2.6)	NS
Dementia, n (%)	21 (33.9)	15 (39.5)	NS
Chronic kidney disease:			
Stage 3a, n (%)	18 (29.0)	19 (50.0)	
Stages 3b-4, n (%)	17 (27.4)	10 (26.4)	0.035 *
Stage 5, n (%)	0	0	NS
Discharge therapy			
Double antithrombotic therapy/triple antithrombotic therapy, n (%)	17/45 (27.4/72.6)	10/28 (26.3/73.7)	NS
Triple antithrombotic therapy duration, n (%)			NS
1 month	29 (46.8)	19 (50)	
3 months	6 (9.7)	3 (7.9)	
6 months	10 (16.1)	4 (10.5)	
12 months	0	2 (5.3)	
Pharmacogenetic study findings			
ABCB1 3435 C>T, n (%)			NS
CC	16 (25.8)	13 (34.2)	
CT + TT	46 (74.2)	25 (65.8)	
ABCB1 rs4148738, n (%)			NS
CC	13 (21.0)	5 (13.2)	
CT + TT	49 (79.0)	33 (86.8)	
CYP3A5*3, n (%)			0.09
GG	54 (87.1)	28 (73.7)	
AG + AA	8 (12.9)	10 (26.3)	
CYP2C19*2 rs4244285, n (%)			NS
GG	43 (69.4)	26 (68.4)	
AA + GA	19 (30.6)	12 (31.6)	
CYP2C19*17 rs12248560, n (%)			NS
CC	36 (58.1)	22 (57.9)	
CT + TT	26 (41.9)	16 (42.1)	
% of platelet inhibition, Me [25;75]	26.5 [7.8;48.5]	22 [8.8.37.3]	NS
RPR (PRU), Me [25;75]	137 [91.5; 181.3]	158.5 [114.3;191.3]	NS
Css min, pg/mL, Me [25;75]	54.8 [35.5;96.3]	60.1 [44.3;128.8]	NS

Css min.—the minimum steady-state concentration of rivaroxaban in blood plasma; RPR—residual platelet reactivity, * *p* < 0.05, NS—not significant.

**Table 5 jpm-13-01371-t005:** Predictors of bleeding based on BARC scale (univariate analysis).

Risk Factors	OR	95% CI	*p*-Value
Genotype TT ABCB1 rs4148738	16.12	3.34–77.88	0.001
Genotype CC ABCB1 3435	3.78	1.12–13.32	0.032
Residual platelet reactivity (PRU < 159)	7.27	1.95–27.13	0.003

**Table 6 jpm-13-01371-t006:** Predictors of bleeding based on the ISTH scale (univariate analysis).

Risk Factors	OR	95% CI	*p*-Value
Css min. of rivaroxaban in blood plasma of more than 137 ng/mL	5.14	1.40–18.87	0.014
Genotype TT ABCB1 rs4148738	7.08	2.17–23.05	0.001
Genotype CC ABCB1 3435	3.13	1.03–9.52	0.044
Residual platelet reactivity (PRU < 159)	2.99	1.02–8.14	0.046

## Data Availability

Certain confidentiality constraints prevent us from disclosing the underlying database used in this study.

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
