# Peer review of "Pharmacokinetic and Pharmacogenetic Predictors of Major Bleeding Events in Patients with an Acute Coronary Syndrome and Atrial Fibrillation Receiving Combined Antithrombotic Therapy"

_jpm, 2023, doi:10.3390/jpm13091371_

Round 1
Reviewer 1 Report
This is very well conducted study assessing bleeding events in patients with AF and ACS focusing on rivaroxaban and clopidogrel.
The conclusions are clear and only specific issues need to be addressed.
· First figure 3 is not clear. It is important to focus only on major bleeding events and it is also important to align ISTH and BARC scores (how many patients are common) in a different figure or more comprehensible.
· Furthermore, results of table 4 and the Kaplan Meier in figure 4 should be based on patients with major bleeding (possibly the 17 patients of ISTH score or more patients with major bleeding if these are identified only by BARC score)
· Another important figure would be a flow chart of patients on DAPT and TAPT (different schemes) and subsequent major bleeding events.
· Minor issue: what is post-infarction cardiosclerosis?
Author Response
Dear Reviewer,
We sincerely appreciate your thorough review of our manuscript and your valuable comments. We have taken your feedback into careful consideration and have made the following revisions to the article:
-
We have added an additional figure depicting the distribution of major bleeding events according to both the BARC and ISTH scales.
-
Regarding the Kaplan Meier analysis in Figure 5, we have decided to present it based on all bleeding events rather than just major bleeding events due to the limited number of events. We apologize for not being able to incorporate your suggestion in this regard.
-
We have also included an additional illustration presenting a flowchart of patients receiving dual and triple antithrombotic therapy and subsequent bleeding events.
- The term "Postinfarction cardiosclerosis" is commonly used in Russia to describe patients who have experienced a myocardial infarction in the past. We have removed this term from the article to improve clarity.
We sincerely apologize for any inconvenience caused and appreciate your understanding. Thank you again for your valuable input and guidance.
Reviewer 2 Report
I have had the opportunity to review the manuscript entitled "Pharmacokinetic and Pharmacogenetic Predictors of Major Bleeding Events in Patients with an Acute Coronary Syndrome and Atrial Fibrillation Receiving Combined Antithrombotic Therapy". I have the following comments to the authors:
1) The paper is not particularly well written. The text is sometimes challenging to understand. The manuscript presentation should be improved. Please, correct typos and grammatical errors in the manuscript.
2) Abstract and Main Text. BARC and ISTH are not "scores" but "classifications" (or scales) for bleeding events. Please revise the manuscript accordingly.
3) Authors should define all abbreviations in the text from the first time they are used. For example, "Cssmax" is not defined.
4) The study was conducted in only 2 Russian centers. Therefore, the generalizability of the results may be limited. The fact that only rivaroxaban was used also limits the generalizability of the results to other DOACs. These aspects should be acknowledged among the limitations.
5) The authors state that "The study enrolled patients with type 2 MI based on the fourth universal definition of MI." Do they mean that they included exclusively patients with type 2 MI? If so, why were other types of MI excluded?
6) "Materials and Methods" section should be divided into sub-headings to improve readability.
7) In the "Results" section, the authors should define the total number of patients included in the study, as well as the number of patients with UA and MI.
8) Table 1. What is the definition of "Postinfarction cardiosclerosis"? Please clarify in the text.
9) The authors should comment on the importance of both the type and duration of triple therapy on bleeding risk in patients with AF undergoing PCI making reference to PMID: 30395219.
10) The authors should better discuss the practical implications of their study.
The paper is not particularly well written. The text is sometimes challenging to understand. The manuscript presentation should be improved. Please, correct typos and grammatical errors in the manuscript.
Author Response
Dear Reviewer,
Thank you for taking the time to review our article. We appreciate your feedback and are committed to improving the quality of our manuscript.
1) We have taken note of your comments regarding the quality of the English language in the manuscript. We have addressed this concern by engaging professional MDPI translators to improve the manuscript's language and readability.
2) The terms "BARC" and "ISTH" have been appropriately referred to as "scales" for bleeding events throughout the abstract and main text.
3) We have carefully reviewed the manuscript and ensured that all abbreviations are defined upon their first usage in the text.
4) Comment has been added as suggested.
5) All types of myocardial infarction were included in the study. We have clarified this in the revised manuscript.
6) Sub-sections have been added as per your suggestion.
7) We have added the total patient count in the "Results" section. MI and UA patient data can be found in Tables 1 and 4.
8) The term "Postinfarction cardiosclerosis" is commonly used in Russia to describe patients who have experienced a myocardial infarction in the past. The term has been removed from the article to improve clarity.
9) We have addressed the importance of both the type and duration of triple therapy on bleeding risk in patients with AF undergoing PCI, referencing PMID: 30395219, in the revised manuscript.
10) We have included a more comprehensive discussion regarding the practical implications of our study.
We sincerely apologize for any inconvenience caused and appreciate your understanding. Thank you again for your valuable input and guidance.
Round 2
Reviewer 1 Report
accepted as it is
Reviewer 2 Report
No further comments.
No further comments.